# Detecting and Understanding the Promotion of Illicit Goods and Services on Twitter

## ABSTRACT

In this study, we reveal, for the first time, popular online social networks (especially Twitter) are being extensively abused by miscreants to promote illicit goods and services of diverse categories. This study is made possible by multiple machine learning tools that are designed to detect and analyze Posts of Illicit Promotion (PIPs) as well as revealing their underlying promotion campaigns. Particularly, we observe that PIPs are prevalent on Twitter, along with extensive visibility on other three popular OSNs including YouTube, Facebook, and TikTok. For instance, applying our PIP hunter to the Twitter platform for 6 months has led to the discovery of 12 million distinct PIPs which are widely distributed in 5 major natural languages and 10 illicit categories, e.g., drugs, data leakage, gambling, and weapon sales. Along the discovery of PIPs are 580K Twitter accounts publishing PIPs as well 37K distinct instant messaging accounts that are embedded in PIPs and serve as next hops of communication with prospective customers. Also, an arms race between Twitter and illicit promotion operators is also observed. Especially, 90% PIPs can survice the first two months since getting published on Twitter, which is likely due to the diverse evasion tactics adopted by miscreants to masquerade PIPs.

## 1 INTRODUCTION

Traditionally, illicit goods and services are promoted either offline or through anonymous online marketplaces [43], e.g., the Silk Road that runs as an onion service. However, these channels tend to have a very constrained audience base [18], and are thus not feasible to promote illicit products to large-scale regular online users. To reach a wider customer base, especially regular online users, alternative promotion techniques have been developed and adopted by miscreants. One typical example is the search engine poisoning attack [27, 38] which involves the injection of illicit promotion into benign but vulnerable websites as well as misleading search engines to index the poisoned webpages with high page rank. However, since a compromised website can be recovered quickly [36], the miscreants have to continuously identify and compromise new websites so as to maintain the magnitude of their promotion campaigns.

Instead, illicit promotion on online social networks (OSNs) is traditionally considered either infeasible or uncommon, since OSNs typically enforce strict content moderation against accounts and posts. However, this is not the case anymore. In this study, we observe that posts of illicit promotion (PIPs) are being distributed at a concerning scale on Twitter, a major online social network platform. Figure 1 presents two PIP examples published on the Twitter platform. One post (Figure1(a)) is intended to promote drug trading, whose attached image shows some hand-written Thai words and several bags of products that look like heroin. And the other post (Figure1(b)) advertises the service of forging certificates and photo IDs. In this post, the main text is masqueraded as a benign English sentence, while both the Chinese username and the attached

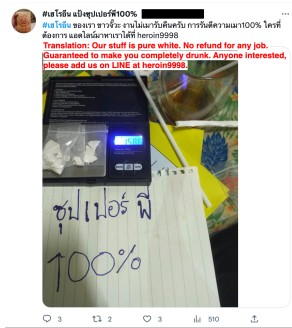
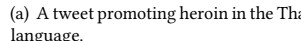

(a) A tweet promoting heroin in the Thai language.

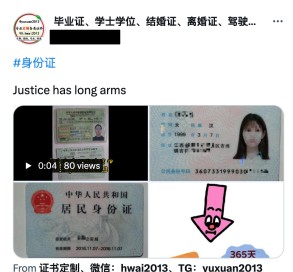

(b) A tweet promoting a certificate/ID forgery service.

**Figure 1: Example posts of illicit promotion on Twitter.**

images clearly promote a fake certificate service. Motivated by such examples, this study aims to gain an in-depth understanding of illicit promotion on Twitter. We focus on the following research questions. First of all, *what illicit goods and services are being promoted on Twitter?* Besides, *how can posts of illicit promotion evade content moderation of Twitter and get published?* Also, *how do the underlying operators of illicit goods and services communicate with the victims (potential customers) as exposed to their PIPs?*

To fulfill these research questions, multiple technical challenges must be tackled. Firstly, given limited access to the Twitter platform, it is challenging to identify PIPs with good coverage. Also, there are no existing tools that can accurately distinguish PIPs from benign Twitter posts (i.e., tweets), not to mention classifying PIPs into well-known categories of illicit goods and services. Furthermore, cybercrime operators tend to embed in PIPs the contacts to facilitate the next-step communication with victims. However, such contacts are of diverse categories and are typically presented in an evasive manner that renders automatic recognition error-prone while keeping them still human-readable. We have addressed these technical challenges through two novel tools. One is the PIP hunter which can not only efficiently search the Twitter platform with known PIP keywords (e.g., relevant hashtags), but also classify whether a given tweet is a PIP or not using machine learning, as well as snowballing this hunting process by automatically generating novel PIP-relevant keywords. The other tool is designed to gain an in-depth understanding of captured PIPs and their underlying campaigns. We thus name it as the PIP analyzer. It consists of a multiclass classifier to classify PIPs into well-defined categories of illicit goods and services, a PIP contact extractor based on named entity recognition (NER), as well as a clustering module designed to group PIPs into their underlying campaigns of illicit goods and services. Leveraging this novel toolchain, we have conducted an

extensive study on illicit promotion on Twitter. Below, we highlight the key findings of this study.

First, *illicit promotion on Twitter features a large scale, diverse categories and products, and a wide distribution across natural languages and Twitter accounts.* Specifically, in total, we have captured 12,401,082 distinct PIPs as well as 580,530 Twitter accounts. These Twitter accounts either publish PIPs (i.e., PIP accounts) or promote illicit goods and services in their personal profiles. Besides, the captured PIPs are widely distributed in multiple natural languages and reside in 10 well-defined categories of illicit goods and services. The top categories with the most PIPs are porn & sex services (69.78%), gambling (13.34%), illegal drugs (8.04%), money-laundry (4.09%) and data leakage(2.03%). Also, from PIPs in each category, a diverse set of specific products have been observed, e.g., methamphetamine and marijuana in the category of illegal drug, and ID cards and passports in the category of data theft and leakage.

Then, *an arms race is observed between illicit promotion campaigns and Twitter content moderation.* On one hand, various evasion techniques have been adopted by PIP operators, e.g., the use of various jargon words, composing PIPs with multilingual characters, and masquerading the tweet text as benign but injecting illicit promotion elements into usernames, media files, or even poll options. The adoption of diverse evasion techniques may explain why over 90% of PIPs can survive the first two months since published. On the other hand, Twitter carries out continuous content moderation and almost 80% of PIPs have got banned six months later after being published, as learned through periodically revisiting captured PIPs and checking their availability.

Furthermore, *for further communication with prospective customers, most PIP operators prefer instant messaging platforms rather than Twitter itself, especially end-to-end encrypted ones.* Especially, we have extracted from PIPs over 37K accounts of five instant messaging platforms including Telegram, WeChat, QQ, WhatsApp, and LINE. Also, miscreants underpinning PIPs of different categories appear to vary a lot in terms of their preferred instant messaging platform, e.g., operators of money laundering or weapon sales prefer Telegram accounts while data leakage services prefer WeChat.

Our contributions can be summarized as below.

• To the best of our knowledge, this is the first extensive security study of illicit promotion on the Twitter platform, which has distilled a set of previously unknown security findings.

• Two novel tools have been designed and implemented to capture and analyze illicit promotion, along with a large dataset of PIPs and PIP contacts captured. We have made both tools and datasets available to foster future research[1].

## 2 BACKGROUND AND RELATED WORKS

**The promotion and communication of the underground economy**. The underground economy, or the shadow economy refers to the production, promotion, trade, and distribution of goods or services that are deemed illicit or even illegal. It ranges from traditional categories (e.g., drug trading and child pornography) to new ones such as hacking services and the trading of illegal data [21]. In many studies, the underground economy is called cybercrime

---
[1]Available in https://illicit-promotion.netlify.app/

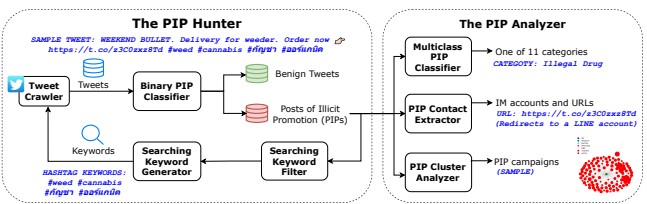

**Figure 2: The methodology to capture and analyze PIPs on Twitter.**

since it has many activities that either have moved to the Internet or owe their existence to the Internet, and we use both terms interchangeably in this study.

A long line of studies has profiled the underground economy from various aspects, among which, a large portion is dedicated to a single cybercrime category, e.g., counterfeit or unlicensed pharmaceuticals [29, 35, 41], drug trading [15, 37], illegal online gambling [22, 46], malware distribution [16, 32, 33, 44], among others. Another line of work examines holistic infrastructures that promote diverse categories of illicit goods and services. Through these studies, various promotion channels have been identified and profiled. To reach a wider customer base, especially regular Internet users, miscreants have also abused or even compromised popular online services. A well-studied example is search engine poisoning attacks [27, 36, 38] (see Appendix A).

Besides, to gain a deep understanding of the underground economy, an important obstacle exists in the use of jargon words among sellers and buyers of illicit goods and services. Some research efforts [47, 50, 52, 54] have thus focused on identifying and understanding such kinds of jargon words. Particularly, Yang et al [47] explored identifying jargon words from search engine keywords as promoted in black hat SEO campaigns, while Yuan et al. [50] detected jargon by analyzing the semantic discrepancy between cybercrime contexts and benign ones for a given word.

**Illicit promotion on OSNs.** Previous works have also made efforts to reveal illicit promotion on OSNs, but most of them focus on a specific category or entity instead of comprehensive detection and understanding. For example, authors of [28, 30] paid attention to illicit pharmacies on Twitter and [51] studied adversarial pornography images crawled from popular OSNs along with the underground business behind them. In this paper, we step forward to build up a general tool chain to detect and analyze PIPs along with efficacy demonstrated for diverse PIP categories and multilingual PIP instances. Furthermore, most findings distilled from our study are applicable to the ecosystem of PIPs rather than being limited to a specific PIP category.

## 3 METHODOLOGY

In this section, we present the methodology to capture and analyze posts of illicit promotion (PIPs) on Twitter. As illustrated in Figure 2, this methodology is comprised of two key modules. One is *the PIP hunter*(§3.1), an automatic pipeline to capture posts of illicit promotion and relevant Twitter accounts. The other is *the PIP analyzer* (§3.2), which is designed to profile PIPs with regards to categories, next-hop contacts, and the underlying campaigns.

**Table 1: The distribution of the PIP ground truth dataset.**

| Language | % Dataset | PIP Category | % Dataset |
|---|---|---|---|
| English | 40.65% | Pornography | 44.47% |
| Chinese | 35.48% | Illegal Drug | 11.45% |
| Japanese | 8.92% | Gambling | 9.50% |
| Thai | 2.64% | Money Laundering | 8.96% |
| Italian | 2.59% | Data Theft and Leakage | 8.85% |
| German | 2.35% | Crowdturfing | 5.10% |
| Spanish | 2.15% | Harassment | 4.08% |
| Russian | 1.81% | Weapon Sales | 2.50% |
| Korean | 1.75% | Forgery and Fake Documents | 2.28% |
| French | 1.65% | Surrogacy | 1.50% |
|  |  | Others | 1.31% |

## 3.1 The PIP Hunter

To hunt PIPs existing on Twitter, a straightforward method is to inspect every tweet, which however requires unlimited access to the Twitter platform. Instead, our PIP hunter is designed to efficiently search Twitter with keywords that are relevant to PIPs. Therefore, our PIP hunter can not only be used by the Twitter platform for internal inspection, but also serve as an effective tool for any third party to audit illicit promotion on Twitter when there is only limited access to Twitter data. At a high level, the PIP hunter consists of a cycle of four steps. To start, it searches Twitter with PIP-relevant keywords, which is followed by a binary PIP classifier that takes a multilingual tweet text as the input and decide whether it is a PIP or not. Given the PIPs identified, the third step is to evaluate the quality of existing PIP keywords and exclude ones with a low PIP hit rate. Then, the last step is to generate keywords from newly captured PIPs and append them to the keyword set so as to boost the next round of PIP hunting. Below, we present more details.

**The tweet crawler.** Given PIP-relevant search keywords, a tweet crawler is deployed to query Twitter using its searching APIs [2] so as to identify tweets and Twitter accounts that are relevant to PIP keywords. Here, the search keywords are either manually crafted in advance or automatically generated by the searching keyword generator introduced soon later. Currently, our tweet crawler supports two types of searching keywords: hashtags and Twitter accounts. For hashtag keywords, the standard Twitter search API will be utilized to retrieve tweets relevant to the given hashtag keyword. Then, when the keyword is a Twitter account, the profile of the account will be retrieved along with its latest tweets up to the crawling time. The number of tweets to retrieve for each account is configurable and should vary across different deployment strategies. In our case, considering the empirical trade-off between collecting more tweets and avoiding repetitive crawling of the same tweet, we set it as 100, i.e., up to 100 latest tweets will be crawled.

**The binary PIP classifier.** Given tweets and Twitter accounts identified through the above searching process, the next step is to distinguish PIPs from benign tweets and benign accounts. As it is observed that the account profile can also be used for illicit promotion, both tweets and account profiles are subject to binary PIP classification. This is achieved through a machine learning classifier which takes either a tweet or the profile of an account

as the raw input, and gives a binary output regarding whether the given content is a PIP or not.

*Classification algorithms.* Three classification options are explored to build up the binary PIP classifier. The first option is a combination of feature embedding through TF-IDF and classification through classic algorithms including SVM and Random Forest. The second option also considers only the text elements of a post. However, it is built up through fine-tuning a transformer-based multilingual language model (bert-base-multi-lingual-cased[4], which has achieved state-of-the-art performance in many text classification tasks [19]. And we name it as *Text-Only Transformer* (see Appendix C for related works on language models). Besides, many posts especially PIPs have media files attached along with the text elements, and these media files (e.g., images or videos) may have visual elements that are important for deciding whether the respective post is a PIP or not. Therefore, the 3rd option we have explored is a multimodal classifier which takes both the visual modality and the text modality into consideration when classifying a post. We thus name it as *Multi-Modality Transformer*. Here, the text modality is encoded using the aforementioned multilingual language model while the visual input is encoded using a pretrained ResNet-152 model [25].

*Labeling the ground truth.* The ground truth dataset is collected through an iterative labeling process involving training weak classifiers, identifying false predictions, and updating the ground truth. Also, across labeling tasks, two labelers independently annotate samples and periodically resolve conflicts. And an inter-labeler agreement rate over 90% is consistently achieved. Please refer to Appendix B.1 to learn more details of the labeling process. The final ground truth dataset consists of 8,408 PIPs and 4,773 non-PIPs that are diverse and representative in categories and natural languages. As shown in Table 1, the dataset are composed in 10 natural languages while the positive samples (PIPs) belong to 11 distinct categories of illicit goods and services.

*Three-fold evaluation.* Our evaluation on the PIP classifiers is three-fold: 5-fold cross-validation upon ground-truth, evaluation on crawled raw tweets, and evaluation on wild tweets that are randomly sampled from Twitter Archiving Project (i.e., wild tweets). After comparing the performance of all classification options, the text-only transformer is selected as the best choice, which consistently achieves a precision of over 94% and a recall of over 96% across all evaluation settings. Please refer to Appendix B.2 to learn the detailed evaluation results.

**Filtering existing PIP keywords**. Some existing PIP keywords may not always work well in terms of triggering new PIPs, a filtering step is thus applied to filter out such ineffective keywords. This is achieved through a threshold-based filtering. Specifically, a metric named as $RCP_{kw}$ is defined as the ratio of new PIPs among all the posts retrieved for a given keyword $kw$ in the current hunting round, and if a keyword has $RCP_{kw}$ lower than a configurable threshold, then, it will be added to a blocklist and will not be used in the future rounds. However, if a blocked keyword hasn't been used for 4 or more rounds but gets extracted again by the keyword generator, it will be unblocked and added back to the keyword set. The threshold of $RCP_{kw}$ has been tuned as 1% in our deployment,

---

[2]https://developer.twitter.com/en/docs/twitter-api

which allows us to gain a good tradeoff between the PIP coverage and the hunting efficiency.

**Searching keyword generator**. The newly captured PIPs will be further fed into the keyword generator so as to extract new keywords which in turn will be appended to the keyword set for the next round of PIP hunting. The search keywords are composed from two sources. One is to extract all the hashtags from identified PIPs, and such seeds are called hashtag keywords. The other is to collect Twitter accounts (users) that either have ever posted PIPs or have their account profiles detected as PIPs, and such seeds are named as account keywords. Many PIP keywords turn out to be effective in terms of discovering previously unknown PIPs, and some can even identify PIPs tha belong to different categories, natural languages, or campaigns. For instance, searching with the Chinese hashtag keyword "广州线下" (GuangZhou In-Person) has identified 16,334 distinct PIPs which are of 5 categories and 10 different natural languages, and have 107 distinct contacts extracted.

**Deployment**. To jump start our PIP hunter, PIP-relevant keywords were first extracted from the thousands of PIPs in the aforementioned ground truth dataset. As PIPs in the ground truth are diverse in categories and languages, so are the derived keywords. Regarding the keyword selection, we also observe that, to capture diverse PIPs with high coverage, the keyword set doesn't have to well cover all categories or languages, nor to be *uniformly* distributed in categories and languages. For example, when starting to compose the ground truth, we only had a keyword set of less than 10 hashtags relevant to drugs and pornography. However, leveraging this small and skewed set, manual execution of the PIP hunting workflow still led to the discovery of PIPs of 10 categories and diverse languages. Multiple factors have contributed to this snowballing effect, e.g., the same hashtag(especially English ones) can be embedded in PIPs of multiple categories and languages.

We then deployed this PIP hunter for the Twitter platform between November 1, 2022 and April 23, 2023. During the deployment, each round started with searching Twitter with keywords, and ended with new PIPs and new keywords discovered. Then, the resulting new keywords will be fed into the next round along with existing ones. Also, to avoid a non-negligible burden to the Twitter servers, our crawler strictly followed the rate limit policies, would suspend the crawling when a rate limit was reached, and would not restart until the rate limit was cleared up. As the hunting process moved forward, the keyword explosion still emerged and conflicted with our limited access to Twitter, e.g., by March 9, 2023, we got 1,280,113 distinct keywords (405,932 hashtags and 874,175 accounts). Therefore, a random sampling strategy was then applied to limit the workload of our Twitter crawler to around 60K PIP keywords. In total, we have scanned over 53 million tweets and discovered 12,401,082 PIPs and 580,530 PIP accounts. Besides, despite being constrained by the limited access to other OSNs, we have also verified the existence and the concerning prevalence of PIPs on the other three OSNs, as detailed in §4.

**The applicability to other OSN platforms.** We also evaluated the PIP hunter on Facebook, YouTube, and TikTok, though only manual assessments were possible due to strict rate limits or lack of APIs. Despite these limitations, we observe that PIP-relevant hashtags collected from one OSN (e.g., Twitter) turn out to be applicable to

**Table 2: Categories of illicit services and goods.**

| Category | Description |
| --- | --- |
| Pornography | Sexual services, and sexually explicit content, e.g., indecent images and videos. |
| Gambling | Online or offline gambling services and products. |
| Illegal Drug | Illegal drugs, e.g., addictive opioid drugs, and prescription drugs. |
| Surrogacy | Surrogacy services, e.g., surrogate motherhood agencies. |
| Harassment | Various harassment services, e.g., cyberbullying, stalking, call/sms bombers. |
| Money Laundering | Various money laundering services, e.g., money muling. |
| Weapon Sales | The sale of weapons, e.g., P99, a semi-automatic pistol and SR-16, a select-fire rifle. |
| Data Theft and Leakage | Services offering stolen sensitive datasets, or various hacking tools and services. |
| Forgery and Fake Documents | Services offering fake or forged documents, e.g., forged passports and fake diplomas. |
| Crowdturfing | Services of illicit crowdsourcing, e.g., deceptive promotion of the popularity of posts or accounts. |

other OSNs, which suggests our hashtag-based searching strategy can likely work for all these OSNs. Besides, when predicting posts collected from OSNs other than Twitter, the binary PIP classifier has achieved a performance that is comparable to that for tweets. We thus believe our PIP hunter is capable of fulfilling the task of cross-OSN PIP hunting, and we leave it as a future work to further explore cross-OSN illicit promotion.

## 3.2 The PIP Analyzer

To gain a deep understanding of PIPs and the underlying promotion campaigns, multiple analysis tools have been built up under the hood of *the PIP analyzer*. These tools include a multi-class classifier to reveal what kinds of illicit goods and services have been promoted in PIPs, a PIP contact extractor to retrieve from PIPs the embedded next hops to communicate with illicit promotion operators, and a PIP cluster to group PIPs into clusters and thus help reveal the campaigns underpinning PIPs.

**The multiclass PIP classifier**. Given identified PIPs, a multiclass PIP classifier is designed to group PIPs into one of ten categories of illicit goods and services. And these categories are learned from aforementioned labeling process and are defined according to previous works[20, 23, 26, 39, 48, 49] and Twitter's policies[11]. The full list of categories is shown in Table 2 along with a short description, while more explanations are presented in Appendix D.1 with regards to the naming and illicitness of these categories.

To build up this classifier, the PIPs in the aforementioned ground truth dataset are reused. Similar to the binary PIP classifier, we explored not only the text-only transformer but also the multimodal transformer. Then, in five-fold cross validation, the multimodal model achieved a precision of 96.86% and a recall of 97.73%, while the text-only model outperformed the multimodal one by 2% in precision and 1.5% in recall. We thus selected the text-only transformer as the default multi-class PIP classifier. For more evaluation results, please refer to Appendix D.2.

**The PIP contact extractor**. Rather than directly communicating with potential victims on Twitter, PIP operators have embedded various contacts in PIPs as next hops for stealthy communication.

These embedded contacts include both website URLs and account IDs of many instant messaging (IM) platforms, which can not only help to gain a better understanding of the underlying campaigns, but also serve as a valuable threat intelligence dataset for future mitigation actions. Thus, a PIP contact extractor is designed to automatically look into a PIP, recognize various contact types, and extract the respective contact entities. Currently, our contact extractor supports the recognition of both websites and accounts of five different IM platforms. These IM platforms include QQ, WeChat, Telegram, Whatsapp, and LINE, which are most frequently embedded in PIPs as observed from our manual study.

Among these 6 contact types, websites can be easily extracted through URL extraction, while WhatsApp and LINE accounts are usually embedded in PIPs as shortened URLs and can thus be recovered by dynamic HTTP visits. For the left three contact types, the extraction is abstracted as a task of multi-class named entity recognition, for which we built up a multi-class classifier to recognize the the BIO tags (beginning, inside, and outside) of these contact types. Please refer to Appendix E to learn more details, e.g., the performance of the NER classifier. Applying this contact extractor to all PIPs revealed over 212K distinct contacts, as detailed in §6.1.

**The PIP cluster analyzer**. Upon PIPs along with respective Twitter accounts and embedded contacts, it is interesting to further uncover the illicit promotion campaigns underpinning PIPs. To achieve this, a cluster analyzer is designed to group PIPs that likely belong to the same underlying campaign. We first abstract PIPs as an undirected graph. In this graph, each PIP account is abstracted as a node of the account type and the size of this node is defined to be proportional to the number of PIPs that this account has posted. Similarly, each PIP contact is also defined as a node but of the contact type. Still the size of a contact node is proportional to the number of PIPs which contain this contact. Then, two nodes will be connected through an edge if they share one or more PIPs. For instance, if a contact node and an account node is connected, there are one or more PIPs that are posted by the account node and contain the entity of the contact node. Similarly, if two contact nodes are directly connected, it means they have been embedded together in one or more PIPs. Given this PIP graph, the flood filling strategy is applied to identify subgraphs isolated to each other. And the resulting subgraphs are considered as separate promotion clusters. The detailed analysis of PIP clusters is presented in §6.2.

### 3.3 Ethical Considerations

Necessary measures have been taken in our study to avoid any potential ethical issues. Particularly, when crawling Twitter for PIP-relevant posts and accounts, our tweet crawler strictly respects the rate limit of the Twitter platform. Then, the collected datasets are securely stored on our research server to which only our researchers have a limited access. Then, when measuring the datasets, we focus on generating statistical data points. When necessary examples are presented, we anonymize any data fields that may leak personally identifiable information.

### 4 POSTS OF ILLICIT PROMOTION

In this section, we profile posts of illicit promotion with regards to their scale, categories, products, distribution, evaision techniques, as well as availability across different online social networks (OSNs).

**Table 3: The ratio of PIPs to the daily Twitter stream at a sampling rate of 1%.**

| Twitter Stream Daily Snapshot | Tweets | % PIPs |
|---|---|---|
| Jan 6, 2023 | 4,142,118 | 3.90% |
| Feb 6, 2023 | 4,119,706 | 2.69% |
| Mar 6, 2023 | 4,044,769 | 3.59% |
| Apr 6, 2023 | 3,973,529 | 4.40% |
| May 6, 2023 | 3,988,035 | 4.75% |
| Jun 6, 2023 | 3,612,604 | 3.10% |

**Table 4: The distribution of PIPs across illicit categories.**

| Category | % PIPs | Category | % PIPs |
|---|---|---|---|
| Pornography | 69.78% | Weapon Sales | 0.58% |
| Gambling | 13.34% | Forgery/Fake Documents | 0.43% |
| Illegal Drug | 8.04% | Others | 0.32% |
| Money Laundering | 4.09% | Crowdturfing | 0.20% |
| Data Theft/Leakage | 2.03% | Harassment | 0.18% |
| Surrogacy | 1.16% | | |

**Scale**. Leveraging both the PIP hunter and the PIP analyzer, by April 23, 2023, we have captured 12,401,082 PIPs in total on the Twitter platform. These PIPs are posted in 5 major natural languages and originate from 580,530 distinct Twitter accounts. Also, 212,689 distinct contact entities have been extracted from these PIPs, which consist of 164,782 URLs (26,831 FQDNs) , 37,621 accounts of 5 different instant messaging platforms, 3,511 Twitter mentions and 6,775 other contacts. One thing to note, due to our limited access to Twitter data, these results can only serve as a lower-bound indicator when estimating the scale of PIPs on Twitter.

**The ratio of PIPs to the Twitter stream.** Another interesting question is to profile what fraction of general tweets are PIPs. As described in §3.1, we utilized daily tweet snapshots from the Twitter Archiving Project to evaluate the generalizability of our binary PIP classifier, which achieves a precision of 99.33%. We then selected the daily tweet snapshot of the 6th day of each month between January 2023 and June 2023, and applied our binary PIP classifier to these 6 snapshots. As presented in Table 3, the ratio of PIPs ranges from 2.69% to 4.75%, which, as a concerning fraction, highlights the prevalence of PIPs on Twitter.

**Categories**. Leveraging the multiclass PIP classifier (§3.2), all the captured PIPs have been predicted into one of well-defined 11 categories of illicit goods and services. The distribution of PIPs across these categories is listed in Table 4. And we can see the top 3 categories are Pornography,Gambling and Illegal Drug, which account for 91.17% PIPs in sum. While more observations are presented in Appendix F due to page limit, one thing that deserves attention is the discovery of a large amount of PIPs promoting products or services of child pornography, which have severely violated Twitter's content policy regarding child sexual exploitation [9]. For instance, Figure3(a) is an example of child porn PIP posted in an evasive manner. The tweet text does not reveal itself as pornography, nor does it specify how to access the pornography resources. Instead, the hashtags and the attached image jointly promote child pornography as well as stating that the Telegram group can be found in the post author's account profile (Figure 3(b)).

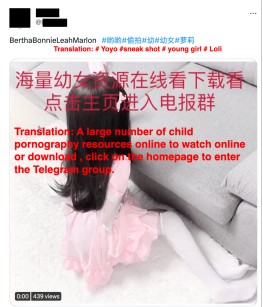
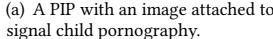

(a) A PIP with an image attached to signal child pornography.

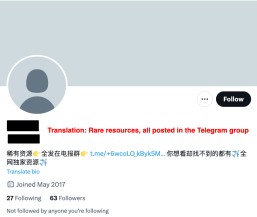

(b) The account profile of the PIP presented in Figure 3(a).

**Figure 3: An example of a pornography PIP.**

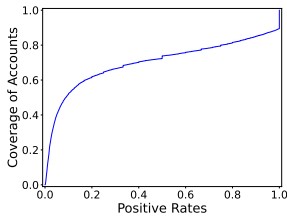
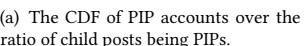

(a) The CDF of PIP accounts over the ratio of child posts being PIPs.

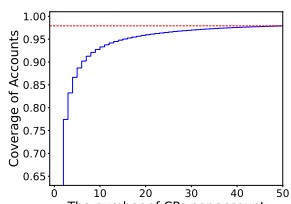

(b) The CDF of PIP accounts against the number of child PIPs.

**Figure 4: The cumulative distribution of PIP accounts.**

**PIP distribution across Twitter accounts.** Given Twitter accounts with one or more PIPs posted, we analyzed their distribution across the PIPs they have posted. We firstly measured the ratio of PIPs over all the tweets an account has posted. Figure 4(a) plots the cumulative distribution of Twitter accounts over their PIP ratio, and we can see it is a highly skewed distribution, with 61.96% PIP accounts have 20% or fewer tweets being PIPs. However, we indeed observed 2,081 PIP accounts whose PIP tweet ratio is higher than 90%. Besides, we also measured PIP accounts over the absolute number of PIPs each of them has posted, which is presented in Figure 4(b). And we can see over 93.30% of accounts posted less than 10 PIPs, while 95.96% accounts have less than 20 PIPs. Furthermore, we ordered the PIP accounts descendingly by the number of PIPs, and the top 1K accounts (0.17%) account for 32.68% PIPs, while it is 68.12% PIPs for the top 10K accounts (1.7% of all PIP accounts). By now, we can conclude that most PIP accounts post a low volume of PIPs while top accounts contribute a large portion of PIPs.

We further investigate the accounts that have all posts detected as PIPs, which comprise 7.87% of all PIP accounts. Our analysis reveals that such accounts tend to be dedicated to promoting a specific category of PIPs. Also, some accounts have posted very few tweets since registration, but rely primarily on their profiles rather than tweets for illicit promotion. Also, among top accounts with the largest volume of PIPs, we observe that a Twitter account can post hundreds of thousands of PIPs without getting banned by the platform. For instance, a PIP account registered in February, 2020

**Table 5: The availability of PIPs on OSNs other than Twitter.**

| Category | Tiktok | Youtube | Facebook |
|---|---|---|---|
| Pornography | 5/20 | 3/20 | 16/20 |
| Gambling | 16/20 | 14/20 | 19/20 |
| Illegal Drug | 7/20 | 1/20 | 19/20 |
| Surrogacy | 8/20 | 5/20 | 15/20 |
| Harassment | 6/20 | 10/20 | 17/20 |
| Money Laundering | 14/20 | 9/20 | 17/20 |
| Weapon Sales | 5/20 | 5/20 | 6/20 |
| Data Theft and Leakage | 16/20 | 4/20 | 19/20 |
| Forgery and Fake Documents | 9/20 | 4/20 | 18/20 |
| Crowdturfing | 19/20 | 17/20 | 20/20 |
| Hit Ratio | 52% | 35% | 83% |

has posted over 198k PIPs in the category of data leakage, which strongly indicates the limitation of Twitter's content moderation.

**PIP distribution across natural languages**. To recognize the natural language of a PIP, *fastText* [3], a library for text representation and classification from Facebook's AI Research (FAIR) lab, is applied. As the result, most PIPs (over 99%) are grouped into 5 major natural languages including Chinese (48.75%), English (38.24%), Japanese (9.98%), Thai (1.70%) and Spanish (0.54%). In terms of illicit categories, the top 5 languages are similar to each other, as Pornography takes a major place in almost all of the top languages, followed by gambling or illegal drug. Considering products of these categories are illegal but can be of great demand in most countries, it's reasonable to have observed such results.

**Social engagement on PIPs.** Over time, PIPs accumulate engagements such as likes, replies, retweets and quotes (i.e. retweets with comments). For our analysis on such social engagement on PIPs, please refer to Appendix I.

> **Finding I**: *Illicit promotion on Twitter is prevalent to a concerning extent, featuring a wide distribution across Twitter accounts, categories of illicit goods and services, and natural languages.*

**Availability across online social networks** To investigate availability of PIPs across other online social network, we conducted a cross-platform analysis for other three major platforms: YouTube, Facebook, and TikTok. For each PIP category, we randomly selected 20 PIP keywords which led to PIPs of the respective category identified on Twitter, and manually searched these platforms for PIPs if any. As shown in Table 5, PIPs are present on all these three platforms, albeit with varying degrees of prevalence. Particularly, Facebook has the highest hit ratio, with 83% of the sampled search keywords having at least one PIP located, and Youtube with the lowest at 35%. Besides, the availability of PIPs varies across platforms and categories. For example, the hit ratios of both *gambling* and *data theft and leakage* are higher on Tiktok than other subcategories. On the other hand, the promotion of *weapon sales* has a low hit ratio on all three platforms, with less than 30% of the sampled seeds having one or more PIPs observed. Interestingly, we found that some PIPs on different platforms share the same contact, indicating that they belong to the same promotion campaign.

> **Finding II**: *Illicit promotion is extensively observed across online social networks, highlighting itself as a cross-platform challenge.*

[3]https://fasttext.cc/

**Table 6: The evasion rate of PIPs across time.**

| Group | Tweeting Period | Evasion Rates | | | |
|-------|-----------------|------|------|------|------|
| | | RV-1 [1] | RV-2 | RV-3 | RV-4 |
| PIP-1 | Oct 24-30, 22 | 21.69% | 21.59% | 21.50% | 21.46% |
| PIP-2 | Oct 31-Nov 6, 22 | 22.33% | 22.25% | 22.12% | 22.08% |
| PIP-3 | Dec 5-11, 22 | 57.78% | 56.41% | 56.02% | 55.87% |
| PIP-4 | Dec 12-18, 22 | 76.18% | 73.08% | 72.93% | 71.26% |
| PIP-5 | Jan 16-22, 23 | 98.27% | 97.62% | 97.62% | 97.62% |
| PIP-6 | Jan 23-29, 23 | 98.16% | 97.90% | 97.90% | 97.11% |
| PIP-7 | Feb 27-Mar 5, 23 | 95.04% | 94.49% | 93.92% | 93.30% |
| PIP-8 | Mar 6-12, 23 | 94.74% | 94.11% | 93.52% | 93.04% |

[1] Revisiting *RV-1* was conducted on April 10-13, 2023, while it is April 15-19, 2023 for RV-2, April 22-26, 2023 for RV-3, and April 29-May 1, 2023 for RV-4.

## 5 CONTENT MODERATION AGAINST ILLICIT PROMOTION

All OSNs under our study claim to have enforced strict content moderation [5, 10, 14], in which case, violative posts should either be blocked from publishing or get unpublished once detected in a later time. However, The existence of so many violative PIPs of diverse illicit goods and services categories suggest respective OSNs especially Twitter fail to prevent PIPs from being posted and becoming visible to OSN users. We thus take a closer look into how PIPs evade the content moderation of Twitter, as detailed below.

**Evasion tactics of PIPs.** To comprehensively analyze the evasion tactics of PIPs, we randomly sample 500 PIPs for each illicit category, manually look into each of them to distll tactics, and implement quantitative analysis on the whole PIP dataset to verify the applicability. The evasion tactics adopted by miscreants can be summarized into three categories: I) Leveraging benign and popular hashtags; II) Using jargon words; III) Embedding illicit promotion messages into any component of a PIP but not its main text. Detailed explanation and examples can be found in Appendix H.

> **Finding III**: *Illicit promotion campaigns have adopted various evasion tactics, likely in an attempt to evade content moderation of the Twitter platform.*

**Content moderation of Twitter.** To further profile the effectiveness of Twitter's content moderation towards published PIPs, we carried out revisiting for captured PIPs. Specifically, 50,000 PIPs hunted in each round are sampled, and their availability are tested by revisiting them periodically (usually at a weekly pace). Given sampled $PIP_{date_p}$ which are first posted on date $date_p$, their *evasion rate* $ER_{date_p, date_r}$ is defined as the ratio of PIPs that are still reachable when being revisited on date $date_r$. Table 6 presents the revisiting results between April 10 and May 1 in 2023, for PIPs posted between Oct 24, 2022 and March 12, 2023. And we can see PIPs posted between Oct 24-30, 2022 have 21.46% still reachable when revisiting 6 month later in the end of April 2023. Also, more than 90% PIPs can survive the first two months since being published.

We then investigate why some PIPs become unavailable during revisits and find out that most are due to account suspension. Specifically, Twitter returns one of six error messages if the PIP is unavailable. These messages can tell us the underlying reasons of PIP unavailability. For instance, a PIP may be unpublished due

to the suspension of the parental account, in which case, the error message will be "*This Tweet is from a suspended account.*". Other reasons include page non-existence, deletion by the author, account non-existence and violation of Twitter's rules. Among unreachable PIPs during revisits, 91.59% are due to account suspension, and 6.22% are due to page non-existence. In summary, we can see that Twitter works in a continuous manner to detect and suspend PIP accounts, resulting in tweets (including PIPs) of the detected PIP accounts also become unpublished.

However, as many PIPs can survive for a long period, we further investigate the difference between unpublished PIPs and surviving ones. To answer this question, PIPs are sampled and divided into two groups depending on the length of their evasion time as observed during revisiting. We then compare both groups with regards to various aspects, e.g., PIP categories, the text syntactic and semantics, the writing style, characteristics of the posting accounts. The only significant difference we have observed resides in the number of PIPs posted by the parental Twitter accounts. For the group of PIPs with a short lifetime, their parental Twitter accounts have 379 PIPs observed on average in our dataset. On the contrary, it is only 62 for the group of PIPs with much longer evasion time. A reasonable explanation is that the more PIPs a Twitter account has posted, the more likely it will be captured and thus suspended by the platform, in which case, all its tweets will also be unavailable, including the PIPs. To further profile this observation, we sampled 153,921 out of all the observed 580,530 PIP accounts, and revisited them during April 15-19, 2023. By then, only 78.95% accounts were still available while all the others got suspended. Comparing alive PIP accounts with suspended ones reveals that alive PIP accounts have 3 PIPs observed on average while blocked ones have 87.

> **Finding IV**: *An arms race is observed between illicit promotion campaigns and Twitter's continuous content moderation, as the result of which, almost 80% PIPs have got banned six months later after being published while on the other hand, 90% PIPs can survive the first two months.*

## 6 CONTACTS AND OPERATORS OF ILLICIT PROMOTION

Given PIPs extensively profiled, we then move the spotlight to the extracted PIP contacts as well as the underlying promotion operators (campaigns).

### 6.1 PIP Contacts

**Scale**. Utilizing our PIP contact extractor, we have successfully extracted a total of 212,689 unique contacts across the Twitter platform from all the PIPs and PIP account profiles. These contacts comprise 12,702 QQ accounts, 11,561 WeChat accounts, 9,644 Telegram accounts, 3,489 LINE accounts, 225 WhatsApp accounts, 3,511 Twitter accounts, 164,782 URLs (corresponding to 26,831 Fully Qualified Domain Names (FQDNs)) and 6,775 other accounts.

Besides, among these contacts, 23.98% are exclusively identified from account profile while 73.92% are only from PIPs and the remaining 2.10% are observed from both account profiles and PIPs. On the other hand, 28.02% PIP accounts have one or more such contacts embedded in either their profiles or their PIPs.

In summary, while promoting illicit goods and services at a large scale on Twitter, the underlying operators prefer platforms other than OSNs for further interaction with their customers, especially instant messaging services. This highlights the necessity and importance of cross-platform collaboration in terms of fighting against illicit promotion activities.

**Distribution**. We have also measured the distribution of contacts across PIPs and PIP accounts, and a major observation is that each contact tends to be promoted through many PIPs and across multiple PIP accounts. Specifically, 5.09% contacts have been embedded into 5 or more PIPs while it is 10 or more for 2.86% contacts.

> **Finding V**: *When it comes to communication with prospective customers, operators of illicit promotion prefer instant messaging platforms and self-managed websites rather than OSN platforms.*

**The adoption of novel IM platforms.** In addition to the widely used communication platforms, we have also observed the adoption of newly emerged end-to-end encrypted communication platforms, such as Wickr[2], BatChat[3], and Potato Chat[7]. All these platforms support end-to-end encrypted communication, just like Telegram and WhatsApp. Furthermore, we find that these secure communication platforms support one or more novel security features which can facilitate more stealthy communication compared with aforementioned well-adopted ones. Specifically, BatChat provides secret chat mode, in which, both parties' avatars are encoded, making it difficult to identify the participants. Additionally, the platform disables the ability to take screenshots or forward chat messages, further safeguarding the content of the conversation. Besides, the secret chat mode of Potato Chat supports even more security features. For instance, Potato enforces message deletion by instructing the recipient's app to delete messages when the sender removes them. Furthermore, users can set self-destruction timers for messages, photos, videos, and files, automatically removing the content from both devices after a specified time.

> **Finding VI**: *Illicit promotion campaigns are increasingly adopting emerging end-to-end encrypted communication platforms, e.g., Wickr, BatChat, and Potato Chat.*

For analysis concerning contact preference across PIP categories and threat alerts from VirusTotal, please refer to Appendix J.

## 6.2 Cybercrime Operators

Through applying the clustering technique introduced in §3.2 to all PIPs captured on Twitter, many PIP clusters have been uncovered, and each cluster consists of PIP tweets, their parental Twitter accounts (PIP authors), as well as contacts embedded in these PIPs. We then conducted manual study for sampled clusters, which confirms that most clusters appear to be separate PIP campaigns. Next, we detail the observations as distilled from analyzing these PIP campaigns.

*Scale of illicit promotion operators.* In total , we have observed 474,979 distinct PIP campaigns. Among these campaign, 93.50% campaigns turn out to be singleton groups and each graph contains only an author node (i.e., a PIP account) and the tweets published by this account. These singleton groups constitute 73.9% PIPs and 76.5% PIP accounts. As for the the remaining 6.5% of the campaigns, 85% are campaigns with only one contact, 8.61% have two contacts, and

3.85% have three or more contacts. With respect to distribution across categories, 87.54% campaigns promote a single PIP category, 9.91% campaigns promote two categories and 2.55% campaigns promote three or more categories. 4 representative clusters are shown in Appendix K.

## 7 DISCUSSION

**Recommendations for real-world PIP mitigation.** We recommend that OSNs should invest more efforts into detecting and removing PIPs from their platforms, particularly for PIPs not in English. PIP detection should not be limited to scrutinizing the text elements, but also take into consideration all elements of an OSN post and its posting account. Also considering PIPs are operated as campaigns, clustering-based detection will be promising and efficient if a low fake alarm rate can be achieved. Also, as demonstrated in our evaluation, the series of tools developed in this study can benefit future endeavors to mitigate illicit promotion in online social networks. Besides, many PIPs prefer IM channels to interact with potential customers while promoting on OSNs, emphasizing the importance of cross-platform collaboration, especially between OSNs and IM platforms.

**Responsible disclosure.** We've been trying to report PIPs and PIP accounts to Twitter, Telegram and other related IM platforms. Up to this writing, LINE responded, assuring us that they're investigating but are unable to provide the results of the investigation. Tencent Security has confirmed and is fixing the issues on QQ and Wechat. For Twitter and Telegram, we reported through both web forms and emails, but have yet to receive any concrete response.

**Code and data release.** We have released the source code of both the PIP hunter and the PIP analyzer, along with the groundtruth datasets to reproduce the machine learning models (e.g., the binary PIP classifiers). In addition, as PIPs may contain illicit content (e.g., child pornography), we release for each illicit category a sampled set of PIPs that have been carefully scrutinized before release. To avoid misuse by miscreants, the full dataset of PIPs and PIP contacts will be provided upon request and background checking. Please refer to the project website (https://illicit-promotion.netlify.app) to access the datasets and source code.

## 8 CONCLUSION

Through this study, we have qualified and quantified the prevalence of diverse posts of illicit promotions (PIPs) on Twitter. Also, it is observed that illicit operators have adopted various evasion tactics when composing and distributing PIPs, which partially explains why so many PIPs could circumvent the content moderation of Twitter, get posted, and keep alive for months before being unpublished. What is also observed is that accounts of instant messaging platforms, especially end-to-end encrypted ones, are frequently used as the next hops for the communication between PIP victims and the underlying illicit operators. Such a cross-platform operation pattern also highlights the importance of security collaborations among OSN and instant messaging platforms for the mitigation of illicit promotion activities.

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

# A  SEARCH POISONING ATTACK

The search poisoning attack is a type of attack wherein the attacker compromises a legitimate website, injects promotional and harmful webpages, induces the search engines to index these webpages with high page ranks, and ultimately expose benign search users to the injected harmful webpages. Such promotional infections have been used to promote diverse cybercrime activities, e.g., online gambling [38], unlicensed pharmacies [36], etc. As revealed by [36], the median time to recover such promotional infections is around 15 days. To detect promotional infections, John et al. [27] proposed the detection of infected webpages by looking at its URL parameters instead of visiting the webpage. Besides, Liao et al. [38] utilized the semantic inconsistency between the injected cybercrime content and the legitimate context of the infected website to decide if a webpage is a promotional infection or not.

# B  THE BINARY PIP CLASSIFIER

## B.1  Ground truth labeling process

An iterative process is followed for labeling, involving 1) searching Twitter with PIP-relevant keywords, which gives crawled tweets; 2) labeling a sampled subset of the crawled tweets to update ground-truth; 3) training a weak PIP classifier using the updated ephemeral ground-truth; 4) applying the weak PIP classifier to predict crawled tweets that are unlabeled, identifying both false positive and false negative predictions; and 5) updating ground-truth accordingly. Besides, when a sample is labeled, it is assigned with not only the binary PIP class, but also one of the PIP categories as listed in Table 2. This iterative process continues until no new PIP categories emerge, each PIP category is well represented in ground-truth and the PIP classifier has achieved a good performance when evaluated on the crawled tweets. Furthermore, inter-rater agreement rate over 90% is achieved across all labeling tasks. Particularly, for 1,000 samples from ground-truth, the agreement rate is 99.8%.

One thing to note, when composing non-PIP posts, we consider only PIP candidates that are not true PIPs, rather than using regular tweets. This is based on the observation that such PIP candidates tend to sit closer to the decision boundary than regular tweets and can help train a more robust PIP classifier. Besides, despite having more PIPs than non-PIPs in the ground truth, we don't observe any negative impact on the binary classifier's performance, which is well demonstrated by the evaluation of wild tweets and crawled

**Table 7: The performance of binary PIP classifiers.**

| Model | Precision | Recall | F1-Score |
|---|---|---|---|
| Text-Only Transformer | 97.25% | 96.04% | 96.64% |
| Multi-Modality Transformer | 96.56% | 97.43% | 97.00% |
| SVM | 95.29% | 95.30% | 95.29% |
| Random Forest | 95.08% | 94.77% | 94.81% |

tweets. Instead, we observe that fewer PIPs can yield comparable performance. For instance, by removing 4K PIPs for balance, a binary PIP classifier trained on this dataset achieved a precision of 96.6% and a recall of 97.4%. More PIPs, on the other hand, help enhance the ground-truth diversity in categories and languages, and are used to train/evaluate the multi-class PIP classifiers as introduced in §3.2.

## B.2  Three-fold evaluation

Our three-fold evaluation consists of 5-fold cross-validation upon ground-truth, evaluation on crawled tweets, and evaluation on wild tweets. Table 7 lists the results of the five-fold cross validation. We can see that the text-only transformer-based classifier has achieved a performance that is comparable to that of the multi-modality one, and in the meantime better than that of the classic classifier. Besides, the multi-modality model has the extra cost of downloading and preprocessing the involved media files, while classic algorithms are inferior to transformer models in terms of data efficiency, multilingual support, automatic feature engineering, and generalizability. We thus choose the text-only transformer model as the default binary PIP classifier.

We further evaluated the selected PIP classifier on unlabeled tweets as collected by our tweet crawler using PIP-relevant keywords. Given prediction results, 500 positive predictions were sampled out for manual validation along with 500 negative predictions. And the manual validation reveals a precision of 94.20% and a recall of 100%.

Then, to further evaluate the generalizability of our binary PIP classifier, we applied it to wild tweets, namely, multiple daily tweet snapshots from the Twitter Archiving Project [4], which archives the tweet stream on a daily basis at a sampling ratio of 1%. Upon predictions on wild tweets, the same manual validation process was applied, which revealed a false positive rate (FPR) of 0.12% with a probability threshold of 0.5. The FRP can be further lowered by tuning the probability threshold. Furthermore, the precision was observed to be 96.8% while the recall is 99.4%. Therefore, considering the low FPR along with the high precision and recall, we can conclude with high confidence that this binary PIP classifier can be used independently to detect PIPs from wild tweets.

# C  LANGUAGE MODELS

A language model [34] is a probability distribution over a set of natural language words, e.g., RNN-based word2vec [42] and transformer-based BERT [19]. A language model is usually trained on a large corpus of unlabeled text documents through self-supervised training tasks such as masked language modeling, i.e., predicting a missing word in a given sentence. Since the hidden layers of such language

---

[4] https://archive.org/details/twitterarchive

models can well represent natural language words in a fixed-size and high-dimensional vector space and are thus commonly used for text encoding, i.e., word embedding. However, training a large and general-purpose language model is both time-consuming and computing-intensive, which motivates the emergence and increasing adoption of the paradigm of pre-training and fine-tuning. In this paradigm, to fulfill an NLP task, rather than building up a DNN model from scratch, a general-purpose and pre-trained language model is first adopted and then fine-tuned on a labeled dataset that is specific to the given NLP task and can be small-scaled. This paradigm outperforms numerous NLP tasks [19, 24, 31]. In this study, to facilitate the detection and understanding of PIPs, we have adopted the paradigm of pre-training and fine-tuning in multiple text classification tasks, e.g., binary PIP classification, multiclass PIP classification, and PIP contact recognition, as detailed in §3. Particularly, across these NLP tasks, we adopt the multilingual BERT [4] as the pre-trained language model, which is built up through two self-supervised training tasks, namely, masked language modeling and next sentence prediction, upon unlabeled Wikipedia corpus which contains numerous text documents written in 102 natural languages.

## D MORE DETAILS ON THE MULTI-CLASS PIP CLASSIFIER

### D.1 The Basis for Defining PIP Categories

For each of the 10 categories, we have observed a reasonable volume of samples when labeling PIPs. We also define another category as *others* to denote PIPs that don't fit in well for the aforementioned categories. Then, regarding the naming of these categories, we try our best to make them self-explained while keeping aligned with relevant terms used in previous works [17, 40, 45, 47, 50, 53]. Particularly, 8 of the 10 categories are considered as illegal in Twitter safety and cybercrime rules [11]. Two exceptions are gambling and surrogacy, likely because they vary significantly in legitimacy across jurisdictions. However, we decide to include them as PIP categories considering multiple factors. Particularly, promotion posts of both categories are often correlated with jurisdictions wherein they are illegal. For instance, 68.29% of surrogacy posts are in Chinese while surrogacy in China is prohibited. Besides, multiple previous studies on illicit promotion also consider both categories [17, 40, 45, 47, 50, 53].

### D.2 More Evaluation Results

For the text-only classification, we still fine-tuned *bert-base-multilingual-cased*[4], a multilingual transformer-based language model. For the multimodal classification, both the text and the visual modality for each PIP have been taken as the input. To build up such a model, *ResNet-152* is used to embed the visual input (i.e., an image), and *bert-base-multilingual-cased* is used to embed the text input. For both models, 80% of the ground truth dataset is used for training, and the remaining 20% is held out for testing. As listed in Table 8, both have achieved very good performance across a set of well-acknowledged metrics. We choose the text-only model as the default multiclass PIP classifier.

**Table 8: The performance of multiclass PIP classifiers.**

| Model | Precision | Recall | F1-Score |
|---|---|---|---|
| Text-Only Transformer | 98.82% | 98.80% | 98.80% |
| Multi-Modality Transformer | 96.86% | 97.73% | 97.27% |

## E THE PIP CONTACT EXTRACTOR

PIP contact extractor is a synthesized tool to extract both website URLs and IM accounts from PIP texts including QQ, Wechat, Telegram, Whatsapp and LINE. Among these contacts, website URLs can be easily identified through regular expression matching. However, many URLs deserve further processing for two factors. On the one hand, some URLs were found to redirect visitors to an IM account, e.g., a Telegram URL *https://t.me/{account_id}*, and we call such URLs as IM URLs. Such IM URLs will be further processed to extract the respective IM type and IM accounts, which is still achieved by defining and applying regular expressions specific to IM platforms. For instance, WhatsApp's IM URL pattern is *https://wa.me/{phone_number}* while it is *https://line.me/ti/p/{accountID}* for LINE. On the other hand, due to the character limit for each tweet, many URLs were found to have been shortened through popular URL shorteners, e.g., *https://bit.ly*, and *https://tinyurl.com*. These shortened URLs will be further visited to recover the true URL of the final landing page. One thing to note, a LINE URL could also be shortened by *http://lin.ee/XXXXXXX*, and it is *http://wa.link/XXXXX* for Whatsapp. Then, after the true URL is recovered, the IM contacts will be extracted.

While LINE and Whatsapp accounts can be extracted from IM URLs, instant messaging accounts of other types are typically embedded into PIPs as ID strings. The account extraction for these contact types is abstracted as a named entity recognition (NER) task. In a nutshell, each word of the PIP text is classified into one of the BIO (beginning, inside, and outside) tags that are specific to each contact type. And contact types under consideration include Wechat, Telegram, QQ, and others. To build up this NER classifier, 3,000 PIPs containing contacts were manually labeled with these tags. In total, the resulting ground truth dataset consists of 386 WeChat accounts, 1,254 Telegram accounts, 626 QQ accounts, and 192 other contacts.

Before training and evaluating the NER classifier, a set of preprocessing steps turn out to be necessary. First of all, as aforementioned, IM accounts can be embedded in PIP as IM URLs (e.g., Telegram URLs). Since these IM URLs can share similar semantic contexts with account IDs of the same IM type, they are likely to be recognized as account ids. Therefore, before classification, an URL in a PIP text will be replaced with a string of *url-x* wherein $x$ denotes its position in the URL list of the same PIP. Also, some PIPs embed various emojis to denote the IM platform, e.g., the use of an airplane emoji (U+2708) in PIPs to denote Telegram. Therefore, a further preprocessing step is to replace emoji symbols with respective natural language descriptions, which was achieved through a Python library *pyemoji*[8].

To train this contact classification model, a multilingual language model, XLM-RoBERTa [13], was fine-tuned with 80% of the ground truth dataset. The resulting model has achieved a micro precision of 95.89% and micro recall of 97.92% when testing on the remaining

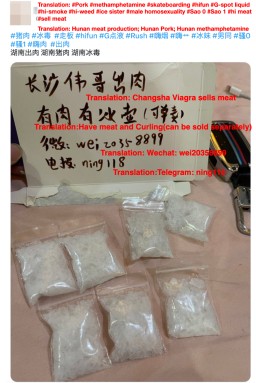

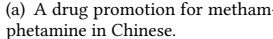

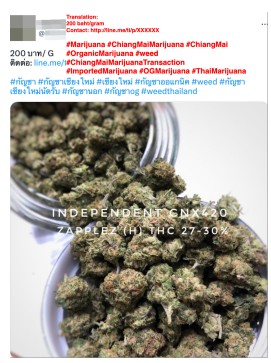

(a) A drug promotion for metham-phetamine in Chinese.

(b) A drug promotion for marijuana in Thai.

**Figure 5: Examples of drug PIPs.**

20% of ground truth. We then applied the contact classifier to all the PIPs, which led to the discovery of 37,621 distinct IM accounts , including 9,644 Telegram accounts, 11,561 WeChat accounts, 12,702 QQ accounts, 225 WhatsApp accounts, and 3,489 LINE accounts. We also manually validated 500 unlabeled PIPs that have IM contacts predicted by the NER model, through which, a precision of 100% and a recall of 86.18% have been observed for this contact extractor.

One thing to note, in some PIPs, contacts are also embedded in images or videos. However, meanwhile, we observe that most of these contacts are also promoted in the text profile of the respective PIP account. As our PIP hunter scans not only tweet contents, but also account profiles, such contacts can still be captured using our text-only classifiers and extractors. To extend coverage to corner cases where the contact is embedded only in images/videos, OCR can be first applied, and we leave it as future work to explore.

## F  PIP CATEGORIES

Among all categories, pornography is the most prevalent one, with 69.78% PIPs belonging to this category. To evade detection, many child pornography PIPs contain only innocent or seemingly harmless text. Instead, the images attached to these tweets contain necessary visual elements that both hint child pornography as well as providing explicit instructions regarding how to access child pornography. Additionally, geographical names are often used in PIPs so as to advertise location-based illegal services, e.g., local sex services. For example, the following two tweets from two different accounts, *"Candice Bridges Elton CopperField #Guangzhou #Guangzhou Massage"* and *"Lesley Browne Baldwin Yerkes #Guangzhou #Guangzhou Massage"*, both use massage and wellness as a way to promote illegal sex services in Guangzhou, China. Also, both differ only in the text but share the same images, hashtags, geographical names, and contacts.

Besides, 8.04% PIPs involve the promotion of various drugs, e.g., methamphetamine, marijuana, and heroin. As illustrated in Figure 5, these tweets promote drugs in Chinese and Thai, respectively. Also, these PIPs have utilized jargon words. For example, "猪肉", a Chinese word denoting pork, is used in illicit promotion to name

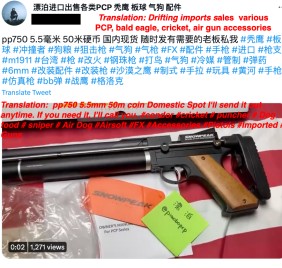

(a) A weapon sale PIP promoting various weapons.

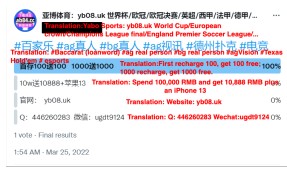

(b) A gambling PIP embedding promotion text and contacts in poll options.

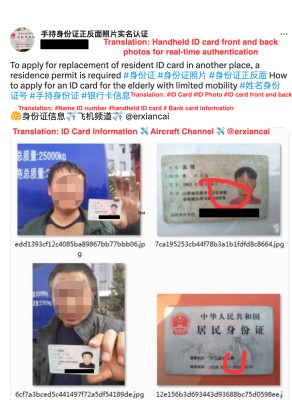

(c) A data leakage PIP selling photo IDs.

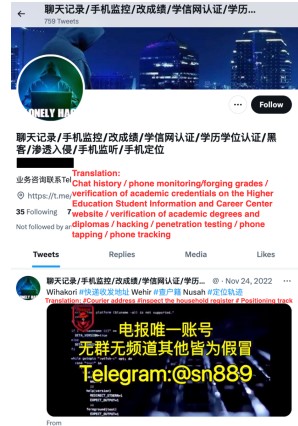

(d) A PIP account providing hacking services.

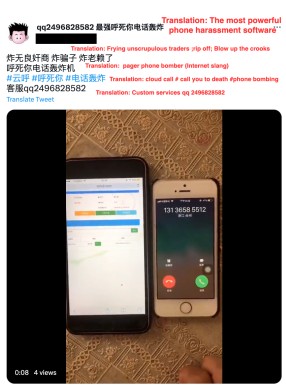

(e) A harassment PIP promoting phone bombing services.

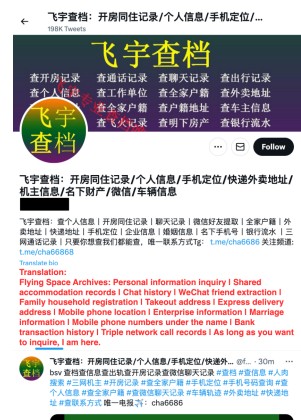

(f) A PIP account posting 198K CPs since registration.

**Figure 6: Examples PIPs of different categories.**

methamphetamine. It is indeed surprising to see them get posted successfully on Twitter since even the tweet text itself appears sufficient to decide it is drug-related, not to mention the hashtags as well as the images. Besides, weapon PIPs often specify the detailed model of the weapons on sale. And some even include images to demonstrate the item's availability. Similarly, harassment PIPs

are mainly used to promote harassment as a service, such as SMS bombing, and phone call bombing. Lastly, PIPs in the category of data theft&leakage encompass many products, such as hacking services, stolen account credentials, unauthorized access to confidential information, and personal data leakage. Figure 6 shows more concrete PIP examples.

## G THE LIST OF JARGON WORDS AS OBSERVED FROM PIPS

The list of examples of jargon words are listed in Table 9 along with descriptions.

## H EVASION TECHNIQUES

Below, we present the different types of evasion techniques that are commonly adopted by miscreants to avoid Twitter content moderation.

**Hashtags.** On one hand, benign and popular hashtags are extensively abused by miscreants to masquerade their PIPs. Across all tweets we've collected, non-PIPs have a median number of 0 hashtags and an average number of 2.27 hashtags, while it is 5 and 6.98 for PIPs respectively, which is much larger than Twitter's official recommendation of one or two relevant hashtags [6]. Specifically, 73.43% PIPs have embedded three or more hashtags while 49.47% have five or more. Besides, PIP operators may exploit hashtags of trending topics or popular events to enhance their tweets' reach. For instance, during the FIFA World Cup 2022, illegal gambling operators were found to inject into their PIPs with football hashtags, #WorldCup, #WorldCup2022, or #WorldCupBetting. Similarly, operators of surrogacy PIPs were found to have embedded hashtags closely relevant to the Mid-Autumn Festival, a popular Chinese festival celebrating family reunion and togetherness. Such hashtags include a Chinese hashtag meaning the Mid Autumn and one more meaning family reunion. In addition to increasing visibility, the injection of benign and popular hashtags into PIPs may likely mislead Twitter's content moderation to some extent. On the other hand, miscreants compose various malicious hashtags so as to keep the main text of a PIP benign while promoting illicit goods and services in the hashtags, e.g., a Chinese hashtag denoting domestic surrogacy, and a Thai hashtag denoting Cannabis Bangkok, and one more Chinese hashtag denoting mobile phone eavesdropping.

**Jargon words.** Across PIPs of different categories, jargon words are commonly used. Particularly, through looking into PIPs, we have manually identified a set of 108 different jargon words that are embedded into 31.21% of all PIPs, which can only serve as a lower-bound estimate for the adoption of jargon words in PIPs. Also, the adoption of jargon words is observed for all the illicit categories. For example, in Chinese PIPs of drugs, "叶子" (leaves in English), or its emoji is used to refer to marijuana, while "猪肉" (pork) and "冰"(ice) and their emojis are used to represent methamphetamine. These terms are derived from either the color or the shape of the respective illegal items. Additionally, a metaphor of the nature of the activity is also popular as jargon words for PIPs. For instance, the farmer and its emoji are used in drug PIPs to refer to people who plant marijuana. The use of jargon words helps illicit promotion operators blend their content with benign tweets, thus

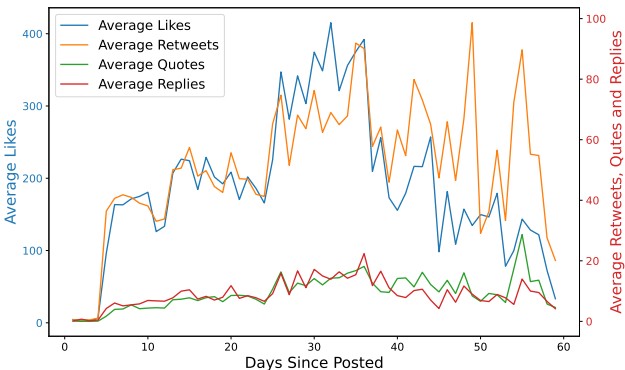

**Figure 7: Average engagements of PIPs over time.**

impeding content moderation. More jargon examples can be found in Appendix G.

**Embedding illicit promotion messages into any component of a PIP but not its main text**. Another evasion pattern we have observed is to embed the promotional text elements into any locations but not the body text of a PIP. Such kinds of locations include attached media files, usernames of the PIP account, description of the PIP account, hashtags, or even poll options.

## I SOCIAL ENGAGEMENT ON PIPS

Over time, PIPs accumulate engagements such as likes, replies, retweets and quotes (i.e. retweets with comments). Given the date a PIP is published $t_p$ and the date it is crawled $t_c$, $t_e = t_c - t_p$ is the elapse time of a PIP, i.e., the days passed since it is posted. As described in §5, 90% PIPs survive the first two months, thus we group PIPs based on elapse time ranging from 1 day to 60 days, and calculate the average engagement of each group. As shown in Figure 7, PIPs indeed receive a non-negligible volume of engagement, e.g., PIPs can receive 374 likes on average after being posted for 30 days, while the average number of likes a regular tweet can receive is 37 [1]. What is also observed is that the extent of engagement declines gradually after 35 days. We believe it is because PIPs with more engagements are more likely to be detected and unpublished by Twitter, which renders the survivorship bias where PIPs with fewer engagements are more likely to survive longer and be counted when calculating engagement of long elapse time. However we are not clear about the reason why average retweets show significant fluctuations after 40 days.

## J ADDITIONAL ANALYSIS ON PIP CONTACTS

### J.1 Evasion techniques for promoting PIP contacts

As discussed in §5, PIP operators have adopted various evasion or promotion tactics so as to make the resulting PIPs appear benign. Furthermore, various evasion techniques have also been observed to hide a contact deeply in the respective PIP. Equipped with such evasion tactics, it would become challenging for the OSN platform to extract the contacts from a detected PIP. For instance, some miscreants use code words, emojis, abbreviations, or unconventional

**Table 9: Examples of jargon words.**

| Word | English | Category | Description |
|---|---|---|---|
| 蓝精灵 | The Smurfs | Drug | Ecstasy |
| 燃料 | Fuel | Drug | The alias of marijuana. |
| 叶子 | Leaves | Drug | Marijuana |
| 飞叶子 | Flying Leaves | Drug | Smoke marijuana |
| 飞行 | Fly | Drug | The Behavior of smoking marijuana. |
| 机长 | Pilot | Drug | Drug dealer selling marijuana. |
| 农夫 | Farmer | Drug | Man who grows marijuana. |
| 鲍鱼 | Abalone | Pornography | Female genitalia |
| 遛鸟 | Walking the birds | Pornography | Men relaxing their genitals in public or outdoors |
| 呦呦 | Yoyo | Pornography | Used to describe child porn. |
| 四件套 | Four pieces set | Data Leakage | Sell the 'four-piece set' of bank cards (mobile phone card, bank card, U-disk, and photocopy of ID card). |
| 跑分 | Benchmarking | Money Laundering | Using a third-party payment account to collect funds on behalf of others, and then transferring the funds to earn a commission. |
| 渔夫 | Fisherman | Money Laundering | Phishing using own personal number, responsible for finding victims |
| 船长 | Captain | Money Laundering | Responsible for setting up phishing sites. |
| 黄河 | Yellow River | Weapon | A brand of gun |
| 海螺 | Conch | Gambling | Board and card game and electronic game system. |

formatting to obscure contact information. Also, contact information of some PIPs are hidden in images or even videos.

**Linguistic obfuscation.** This tactic involves using code words, emojis, abbreviations, or unconventional formatting to obscure contact information. From the extracted contacts in our dataset, we randomly sampled 200 tweets to further investigate the extent of linguistic obfuscation used by illicit operators. A striking 59% of samples employed various linguistic obfuscation techniques. For instance, "QQ" is often replaced with "企鹅", "扣扣", or the emoji Penguin( U+1F427), while "Wechat" is replaced with "薇", "V", "VX", the heavy black heart (U+2764), or the satellite emoji (U+1F6F0). Similarly, "Telegram" is often substituted with "airplane" or the airplane symbol (U+2708), and "Line" with "赖". These substitutions are often based on similar pronunciations (e.g., "Line" in Chinese sounds like "赖", U+1F6F0 and U+2764 are similar to "WeChat" in Chinese) or resemblances to the application's icons (e.g., the Telegram icon is an airplane(U+2708), and the QQ icon is a penguin (U+1F427)).

**Visual representation and steganography.** The visual representation of contacts involves presenting contact information as visual elements in images or even videos. This can be further strengthened by the use of human-written text (Figure 5(a)), which makes it difficult for even OCR systems to recognize and extract the respective contact information. Embedding contact information within seemingly innocuous elements is also a way of stealthy promotion, e.g., embedding the contact in the poll options instead of the tweet's main text. Additionally, illicit operators may also embed contact information within any frames of innocuous videos. Currently, our PIP contact extractor supports only the retrieval of contacts from text inputs, namely, the PIP text and the profile text of a PIP account. We leave it as our future work to extract contacts from media files and other seemingly innocuous components of a PIP.

**Private messaging.** By requesting potential buyers to send private messages for more information (e.g., Figure 6(a) in Appendix ??), illicit operators can avoid the direct exposure of their contact information in OSNs. This tactic enables them to selectively respond to

**Table 10: The distribution of most preferred contact types for PIPs in different categories.**

| Category[1] | Telegram | Wechat | QQ | LINE | WA[2] |
|---|---|---|---|---|---|
| Pornography | 43.6% | 30.24% | 34.02% | 2.49% | 0.02% |
| Illegal Drug | 36.99% | 24.91% | 13.79% | 14.73% | 1.12% |
| Gambling | 42.45% | 41.39% | 24.73% | 2.96% | 0.97% |
| Surrogacy | 22.92% | 67.76% | 14.26% | 0.03% | 1.38% |
| Harassment | 25.60% | 22.78% | 56.23% | 0.01% | 0.00% |
| Money Laundering | 68.17% | 19.77% | 17.91% | 0.04% | 0.24% |
| Weapon Sales | 73.36% | 18.80% | 7.86% | 0.1% | 2.5% |
| DTL [2] | 35.75% | 58.91% | 11.51% | 0.02% | 1.50% |
| FFD [2] | 16.64% | 68.76% | 14.81% | 0.47% | 0.00% |
| Crowdturfing | 55.23% | 44.18% | 11.04% | 0.00% | 0.00% |

[1] Note that one PIP may include more than one contact.
[2] WA denotes WhatsApp, FFD is short for Forgery and Fake Documents and DTL is short for Data Theft and Leakage.

potential buyers and increase the effectiveness of their promotional efforts.

## J.2 Contact preference across PIP categories

Regarding contact selection, illicit promotion campaigns of different categories exhibit distinct preferences. Table 10 presents, for each PIP category, how the extracted contact entities distribute across well-known contact types. *Porn, money laundering,* and *weapon sales* predominantly prefer Telegram accounts, which are associated with 43.60%, 68.17%, 73.36% of PIPs of the corresponding category, respectively. For *harassment* activities, QQ emerges as the most favored contact method, accounting for 56.23% of harassment PIPs. Besides, for *data theft & leakage* and *fake documents*, WeChat is much more popular than others.

## J.3 Threat alerts from VirusTotal

To profile the threats of websites embedded in PIPs, we retrieved and analyzed their threat reports from VirusTotal [12], a widely recognized open threat exchange platform. Due to the rate limit of

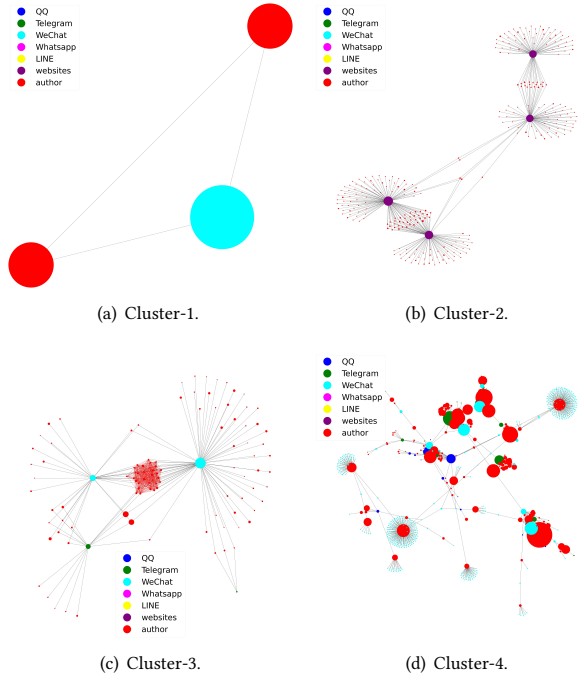

| (a) Cluster-1. | (b) Cluster-2. |

| (c) Cluster-3. | (d) Cluster-4. |

**Figure 8: Representative 4 PIP clusters.**

VT, we only randomly sampled 20k PIP website URLs (corresponding to 7,391 FQDNs) for analysis. As shown in Table 11, 0.80% URLs have one or more alarms triggered, while it is 2.46% for FQDNs. Furthermore, among these alarmed URLs, 11.38% are considered malware websites, while 24.39% are considered phishing websites.

**Table 11: Statistics of VirusTotal reports for PIP websites.**

| Category | Count | Reported | Alarmed | Malware | Phishing |
|---|---|---|---|---|---|
| URLs | 20,000 | 7.88% | 0.80% | 0.07% | 0.15% |
| FQDNs | 7,391 | 69.30% | 2.46 % | 0.32% | 0.70% |

## K REPRESENTATIVE CAMPAIGNS UNDERPINNING PIPS

we looked further into the campaigns and discovered some commonly used schemes in cybercrime promotion. In general, the most notable characteristic of the campaigns is that most operators take control of at least two accounts to accomplish promotion and these accounts are connected by common contacts or websites. We present four campaigns as below under the name of Cluster-1 to Cluster-4.

*Cluster-1.* Figure8(a) visualizes cluster-1 as a graph wherein nodes denote either PIP accounts or contacts, and the size of each node represents the number of PIPs it is associated with. As shown in Figure8(a), cluster-1 consists of 3 nodes (two account nodes and one WeChat contact), 3 edges, and 6,328 PIPs, and all PIPs are classified as data-leakage. The two Twitter accounts have published similar amounts of PIPs and promoted the same WeChat contact, and are very likely to be operated by the same person or organization.

*Cluster-2.* As shown in Figure8(b), cluster-2 consists of 290 nodes and 349 edges, containing 412 PIPs all classified as pornography. Different from Cluster-1, Cluster-2 is composed of much more small-scale accounts connected to four fully qualified domain names, which are *voice-live.liblo.jp*, *live-video.golog.jp*, *live-video.liblo.jp*, and *video-liv-e.liblo.jp*. By looking up DNS resolutions for the FQDNs, we discovered that they share the same server IP (147.92.146.242) of a porn website. Therefore, it's reasonable to infer that Cluster-2 is operated in an organized way by the same operator of the website.

*Cluster-3.* As shown in Figure8(c), Cluster-3 consists 106 nodes, 125 edges and 516 PIPs, 97.10% of which are classified as fake document. It's worth noting that a subset of author-type nodes are connected by red edges and form a fully connected component in the middle of Figure8(c), which means that these accounts have common contacts embedded in their user profiles. As mentioned before, embedding contacts in components other than main text of the tweet is a common evasion tactic utilized by miscreants.

*Cluster-4.* As shown in Figure8(d), Cluster-4 has 708 nodes and 887 edges, associated with 702 PIPs. 99.02% of PIPs are classified into the category of data-leakage. Nodes of Cluster-4 consists 163 authors, 29 QQ contacts, 21 Telegram contacts and 495 Wechat contacts. Unlike Cluster-1 to Cluster-3 which mainly use a single kind of contact, diversity in contact types but still high concentration on single category of illicit goods and services makes cluster-4 a special one.

