# OpenReview forum: "Detecting and Understanding the Promotion of Illicit Goods and Services on Twitter"
_ACM.org/TheWebConf/2025/Conference — WWW 2025 Poster_

### Official Review · Reviewer_JMem · 2024-11-13

**Novelty:** 5
**Technical Quality:** 6

**Review:**

Summary of the Paper: This paper aims to deeply understand the promotion of illicit goods on Twitter by examining various aspects, including scale, categories, distribution across Twitter accounts, distribution across natural languages, content moderation, and the underlying contacts and operators involved in illicit goods promotion. The authors analyzed a large dataset of illicit goods promotion posts collected over six months. In total, they scanned over 53 million tweets and discovered 12,401,082 posts promoting illicit goods as well as 580,530 accounts engaged in this activity. The authors also designed and implemented two novel tools to search, classify, and analyze illicit promotion posts. The findings indicate that posts promoting illicit goods have been observed on Twitter and are common across other online social networks. These posts encompass various categories such as drugs, data leakage, gambling, and weapons sales. They have circulated on Twitter using various evasion tactics, including: I) leveraging benign and popular hashtags; II) using jargon; and III) embedding illicit messages. Some posts are suspended after two months, while others continue to circulate for six months. Communication with consumers of these illicit promotions can persist through self-managed websites and end-to-end encrypted communication platforms. The implications of this work highlight the widespread distribution of illicit goods promotion on online social media platforms and call for greater efforts from these platforms to detect and remove such posts.

Pros: The research topic of understanding the various aspects of illicit goods promotion on online social networks is important and timely. The dataset is substantial, covering six months of Twitter posts. The methodology for capturing and analyzing posts promoting illicit goods is well-designed and implemented. To gain a deeper understanding of this promotion, the novel tool consists of two modules: the PIP Hunter, an automated pipeline to capture posts of illicit promotion and relevant Twitter accounts, and the PIP Analyzer, a multi-class classifier that profiles and categorizes the types of illicit goods and services being promoted. The tools are built on a multimodal classifier that considers both visual and textual modalities when classifying a post, providing a deeper understanding of the nature of the content. The analysis of evasion tactics used to circulate these posts, as well as the survival period of the posts, is particularly interesting. The findings are well-documented, featuring clear tables, plots, summary text boxes, and appendices.

Cons: The categories of illicit services and goods summarized in Table 2 have vague or unclear definitions. These categories could be more thoroughly defined and could benefit from the inclusion of sub-categories (e.g., cyberbullying, stalking) for illicit services and goods posts. The implications section is written in a superficial manner and does not provide actionable insights for addressing the problem. Additionally, there is no discussion of limitations or future work. Future research could extend the language analysis to include a geographical analysis of the locations of tweets.

**Questions:**

1-	Are the authors planning to extend this work or narrow it down to include further analysis on sub-categories (such as cyberbullying and stalking) of illicit services and goods posts?
2-	Will future work include an analysis of illicit services and goods posts on other social media platforms?

**Ethics Review Description:**

The authors included a Section on Ethical Considerations (Section 3.3), which is I found satisfactory.

**Reviewer Confidence:**

3: The reviewer is confident but not certain that the evaluation is correct

**Scope:**

4: The work is relevant to the Web and to the track, and is of broad interest to the community

---

### Official Review · Reviewer_pam9 · 2024-11-24

**Novelty:** 3
**Technical Quality:** 3

**Review:**

__Pros__

* The paper offers valuable insights into the landscape of PIPs on Twitter, highlighting their distribution across various categories and languages. This comprehensive analysis provides a clearer understanding of how PIPs manifest and operate within the platform, adding depth to the study's findings.

* The system is platform-agnostic, meaning it can be applied to identify PIPs across various online social networks (OSNs). This flexibility enhances its utility, allowing it to adapt to different platforms and broaden its impact in detecting and addressing PIPs.

__Cons__

* Internally, PIPHunter utilizes a binary PIP classifier that analyzes either a tweet or an account profile to determine whether the content is classified as PIP or not. In my view, this classifier represents one of the paper's most valuable contributions, as it could be seamlessly integrated into automated systems to proactively detect and remove PIP content without manual intervention. However, the paper lacks any evaluation metrics to assess how effectively PIPHunter distinguishes between benign and PIP content. While I understand that the paper’s primary focus is on measurements, understanding the accuracy and reliability of the underlying tools is equally critical. Including such evaluation metrics would greatly enhance the credibility and practical relevance of the proposed system.

NOTE: I see that the precision is claimed to be 99.33% in section 4, but these metrics need their own section to discuss how the system was evaluated.

* PIPHunter is not compared to any existing systems or related work, which makes it challenging to evaluate the paper's novel technical contributions. Without a clear comparison, it's difficult to understand how PIPHunter stands out or improves upon prior methods in the field. Including such comparisons would provide valuable context and better highlight the system's unique strengths and innovations.



Overall, the paper's primary contributions appear to lie in its role as a measurement study. While its findings provide some scientific value, they are not particularly surprising or groundbreaking. The outcomes of the measurements, while informative, lack the excitement or novelty that might make them more impactful.

A notable drawback is that the measurement data feels dated, which diminishes the relevance of the insights. It’s disappointing that the authors did not choose to analyze more recent datasets, as doing so could have enhanced the paper’s significance and ensured its findings reflect the current landscape.

Finally, the paper would benefit greatly from a more thorough evaluation of PIPHunter's technical capabilities, particularly the accuracy of its classifiers. Metrics such as accuracy, precision, recall, and F1 score are essential to assess its effectiveness in identifying PIPs. Additionally, a comparison with existing systems is crucial to demonstrate the technical superiority of PIPHunter's approaches. Without these evaluations, it is difficult to fully understand or validate the system’s performance and its contributions to the field.

**Questions:**

1. On page 2, the authors mention that "90% of PIPs can survive the first two months since published. On the other hand, Twigger carries out continuous content moderation and almost 80% of PIPs have got banned six months later after being published" Do you all know why this is the case, what prevents Twitter from catching this in a timely manner?

2. The paper states, "when the keyword is a Twitter account, the profile of the account will be retrieved along with its latest tweets up to the crawling time." Does this require an exact match to the account name? How are you all querying the entire set of accounts? How you all search for hashtag is clear, but how you all do the account querying is not.

3. The dataset is quite old  (1.5 years since the crawling stopped). Have you all looked at any more recently crawled datasets?

4. When discussing PIPs across other online social networks, what is defined as the "hit" ratio?

**Ethics Review Description:**

The paper has a non-nude CSAM victim in Figure 3.a. I am not sure what the policy around this is (i.e., should this image be removed or blurred to protect the victim), so I am flagging it as an ethics issue.

**Ethics Review Flag:**

Yes

**Reviewer Confidence:**

3: The reviewer is confident but not certain that the evaluation is correct

**Scope:**

3: The work is somewhat relevant to the Web and to the track, and is of narrow interest to a sub-community

---

### Official Review · Reviewer_3psN · 2024-11-26

**Novelty:** 5
**Technical Quality:** 4

**Review:**

Summary
This paper proposes that miscreants extensively abuse popular online social networks (especially Twitter) to promote illicit goods and services of diverse categories.  This study is made possible by multiple machine learning tools that are designed to detect and analyze Posts of Illicit Promotion  (PIPs) as well as revealing their underlying promotion campaigns.

Strengths
1. Relevance and Timeliness: The study addresses a critical issue of illicit goods and services promotion on social media, a growing concern for public safety and regulatory authorities.
2. Large-scale Data: The analysis leverages a vast amount of data (12 million PIPs), which strengthens the study’s validity and provides a broad overview of the scale of the issue.
3. Multilingual and Multi-platform Scope: The research is comprehensive, analyzing content across five major languages and several social networks, providing a global perspective on illicit promotion.

Weaknesses
1. Lack of Clear Actionable Solutions: While the study uncovers the scale and tactics of illicit promotion, it offers limited discussion on practical solutions for mitigating or preventing the issue.
2. Survivability of PIPs: The observation that 90% of PIPs survive for over two months due to evasion tactics suggests that the current detection methods may not be entirely effective. More emphasis could be placed on improving detection systems.
3. Ethical and Legal Considerations: The paper may raise concerns about privacy, data collection, and potential misuse of the information, though these aspects are not discussed in detail.

**Questions:**

What is the accuracy rate of the machine learning tools in detecting illicit posts, and how do you handle false positives (e.g., legitimate content flagged as illicit)? Could you provide any insights into the balance between precision and recall in your model and how you address the trade-off?

**Reviewer Confidence:**

3: The reviewer is confident but not certain that the evaluation is correct

**Scope:**

4: The work is relevant to the Web and to the track, and is of broad interest to the community

---

### Official Review · Reviewer_5k4N · 2024-11-29

**Novelty:** 2
**Technical Quality:** 2

**Review:**

- Thank you for submitted to WWW 2025. This paper analyzes illicit promotion activities on Twitter by developing machine learning tools to detect and analyze Posts of Illicit Promotion (PIPs). While the work presents some interesting findings about the scale and nature of illicit content promotion on social media platforms, I have several significant concerns about this study's significance, methodology, and potential impact.

- First, the fundamental research motivation and problem statement could be more robust. The authors observe that illicit promotion exists on Twitter and set out to measure it without clearly articulating why this is an important security problem that warrants dedicated research attention. The paper lacks a compelling threat model or demonstration of the actual harm caused by these promotional posts. Many of the "illicit" categories studied, like gambling and adult content, exist in legally gray areas that vary by jurisdiction.

- The methodology has significant limitations. The authors rely heavily on keyword-based crawling, which introduces inherent sampling bias. The ground truth labeling process is poorly documented - how did the authors determine what constitutes "illicit" content? The evaluation of the ML classifiers focuses primarily on standard metrics like precision/recall without analyzing failure modes or conducting ablation studies to understand what features drive performance.

- The paper's technical contribution is limited. The ML techniques used are relatively standard applications of BERT and other common approaches. The contact extraction and clustering methods are straightforward applications of existing techniques. No significant algorithmic or methodological innovations are presented.

- Most concerningly, the results section focuses heavily on descriptive statistics without deep insights into the underlying phenomena. Primary findings like "porn is common" and "operators use evasion tactics" are unsurprising. The analysis of promotion campaigns and operator behaviors needs to be more superficial.

**Questions:**

- Given the lack of concrete evidence of harm from illicit promotion on Twitter, how do the authors justify the significance of the research? What specific security threats does this activity pose?

- The ground truth labeling process appears biased and arbitrary. How did authors systematically determine what content qualified as "illicit" versus legitimate promotion?

- The paper needs a more rigorous analysis of potential negative impacts. How could these detection tools be misused for censorship or surveillance? What safeguards have been considered?

- The evaluation focuses mainly on standard ML metrics. What analysis of failure modes have you done? How robust are the classifiers to adaptive adversaries?

- Many of the authors' findings seem obvious or previously known. What novel insights does this work provide beyond confirming existing understanding of social media misuse?

**Reviewer Confidence:**

3: The reviewer is confident but not certain that the evaluation is correct

**Scope:**

3: The work is somewhat relevant to the Web and to the track, and is of narrow interest to a sub-community

---

### Official Review · Reviewer_zPwN · 2024-12-02

**Novelty:** 4
**Technical Quality:** 5

**Review:**

Pros

- New insights about how moderation works in social networks.

Cons

- Unclear how the corpus of original keywords was derived.

- FP/FN analysis.

- Hard to assess the delta.

Details

Overall, this is a nice study on how illegal content can escape moderation in a
popular social network, such as Twitter.com (now X.com). However, there are
several parts in the paper, which hide necessary technical details.

- Initial seed. The authors start with a set of keywords for discovering tweets
  with illegal content. It is unclear how this first seed of keywords was
  derived.  The corpus of the keywords is further augmented, however, the
  strategy of deriving/extending these keywords is not thoroughly discussed in
  the paper.

- FPs/FNs. I think one of the major weaknesses of the paper is the complete
  absence of discussion about false positives/negatives, especially when the
  identified tweets are over 12 millions. It is unclear if these identified
  tweets are not misclassified or, in other words, what is the confidence level
  of the classification.

- Delta. There is no comparison with past studies for quantifying the increase
  of tweets with illegal content. The level of tweets with illegal content, as
  reported in the paper, might is alarming according to the authors. However,
  how can we be sure about this, when there is no comparison with the past?
  Moreover, there is no attribution about this level change, if it is indeed
  happening.

**Questions:**

- How did you compute the initial set of keywords?

- How do you evaluated your FPs/FNs in your tweet classification?

**Reviewer Confidence:**

3: The reviewer is confident but not certain that the evaluation is correct

**Scope:**

4: The work is relevant to the Web and to the track, and is of broad interest to the community